# Anti-Fibrotic and Anti-Angiogenic Activities of *Osbeckia octandra* Leaf Extracts in Thioacetamide-Induced Experimental Liver Cirrhosis

**DOI:** 10.3390/molecules26164836

**Published:** 2021-08-10

**Authors:** Sudarma Bogahawaththa, Suranga P. Kodithuwakku, Kavindra K. Wijesundera, Eranga H. Siriweera, Lalith Jayasinghe, Waruna L. Dissanayaka, Jayanthe Rajapakse, Chandana B. Herath, Tadayuki Tsujita, Missaka P. B. Wijayagunawardane

**Affiliations:** 1Department of Animal Science, Faculty of Agriculture, University of Peradeniya, Peradeniya 20400, Sri Lanka; 20974007@edu.cc.saga-u.ac.jp (S.B.); surangap@agri.pdn.ac.lk (S.P.K.); 2The United Graduate School of Agricultural Sciences, Kagoshima University, Kagoshima 890-0065, Japan; 3Laboratory of Biochemistry, Faculty of Agriculture, Saga University, Saga 840-8502, Japan; 4Department of Veterinary Pathobiology, Faculty of Veterinary Medicine and Animal Science, University of Peradeniya, Peradeniya 20400, Sri Lanka; kavindra77@gmail.com (K.K.W.); jayanthar@pdn.ac.lk (J.R.); 5Department of Pathology, Faculty of Medicine, University of Peradeniya, Peradeniya 20400, Sri Lanka; eranga.siriweera@gmail.com; 6National Institute of Fundamental Studies, Hanthana Road, Kandy 20000, Sri Lanka; lalith.ja@ifs.ac.lk; 7Applied Oral Sciences & Community Dental Care, Faculty of Dentistry, The University of Hong Kong, Hong Kong SAR, China; warunad@hku.hk; 8Department of Medicine, Austin Health, Melbourne Medical School, The University of Melbourne, Heidelberg, VIC 3084, Australia; CHerath@unimelb.edu.au; 9South Western Sydney Clinical School and Ingham Institute for Applied Medical Research, Faculty of Medicine, University of New South Wales, Liverpool, NSW 2170, Australia

**Keywords:** hepatoprotective, anti-angiogenic effect, *Osbeckia octandra*, Wistar rats, thioacetamide, cirrhosis

## Abstract

Chronic liver inflammation has become a major global health concern. In the absence of clinical surrogate markers to diagnose inflammatory liver disease, the intervention with effective drugs in modern medicine tends to be late. In Sri Lanka, traditional medical practitioners prescribe herbal preparations from *Osbeckia octandra* for the prevention and treatment of liver disorders. To test the efficacy of such treatments, we have administered thioacetamide (TAA) to male Wistar rats to induce chronic liver damage (disease control; DC) and examined how various leaf extracts: crude leaf suspension (CLS), boiled leaf extract (BLE), sonicated leaf extract (SLE), methanol leaf extract (MLE) and hexane leaf extract (HLE) of *O. octandra* ameliorate TAA-induced liver disease. The CLS, BLE and SLE treatments in cirrhotic rats significantly attenuated disease-related changes, such as liver weight and hepato-enzymes. The mRNA levels of *Tnf-α* were significantly decreased by 3.6, 10 and 3.9 times in CLS, BLE and SLE compared to DC. The same treatments resulted in significantly lower (19.5, 4.2 and 2.4 times) *α-Sma* levels compared to DC. In addition, *Tgf-β1* and *Vegf-R2* mRNA expressions were significantly lower with the treatments. Moreover, BLE expressed a strong anti-angiogenic effect. We conclude that CLS, BLE and SLE from *O. octandra* have potent hepatic anti-fibrotic effects in TAA-induced liver cirrhosis.

## 1. Introduction

The liver is an important organ that contributes to the metabolism of carbohydrates, amino acids and lipids. It is also involved in the detoxification and elimination of endogenously and exogenously derived toxic substances [1]. Toxic insults or microorganism infection can injure liver parenchymal cells, subsequently leading to the aberration of liver architecture and abnormal metabolic functions, which in turn results in tissue inflammation, fibrosis and total liver failure [2,3]. Liver cirrhosis affects more than 2% of the world’s population and is responsible for about one million deaths annually due to a lack of effective medical therapies [4].

Owing to the lack of sensitive surrogate markers to diagnose chronic liver injury, intervention timing tends to be late. This leads to progressive liver damage up to severe liver cirrhosis. The only treatment option for this stage to date is liver transplantation; however, this is hampered by a lack of donors. As a result, there remains a major need to identify potentially modifiable factors that exacerbate the liver injury and fibrosis and to develop therapies that can prevent or slow down liver scarring. The treatment of liver disease with plant extracts is cost-effective, easily accessible and has less or no side effects [5]. *O.*
*octandra* (*Heen Bovitiya*) is an indigenous herb [6] found in Sri Lanka that has traditionally been used to treat liver diseases [7]. Previous work has suggested that natural antioxidants present in the extracts of *O. octandra* can overcome oxidative damage caused by reactive oxygen species of the injured liver with the arrest of subsequent development of liver fibrosis [8].

Furthermore, studies conducted in vivo [9] demonstrated that an aqueous extract of *O. octandra* leaves could protect against acute hepatotoxicity induced by carbon tetrachloride (CCl_4_) in rats and paracetamol-induced liver injury in mice [6]. Moreover, *O. octandra* leaf extracts have been shown to possess protective properties on d-galactosamine- and *tert*-butyl hydroperoxide (TBH)-induced toxicity in isolated rat hepatocytes [9].

We, therefore, investigated the hepatoprotective effects of *Osbeckia octandra* leaf extracts in thioacetamide (TAA) induced cirrhotic rats, as pathological lesions observed in this model resemble those that can be seen in human cirrhosis [10]. Moreover, the molecular events imparting hepatoprotective effects of *O. octandra* have not been studied and reported previously. The highly vascularized fibrous tissue that surrounds the regenerative hepatic nodules suggests that angiogenic factors may be involved in the pathogenesis of the disease. Therefore, in the present study, we also investigated the anti-angiogenic properties of *O. octandra* in the (human umbilical vain endothelial cells (HUVEC) model as well.

## 2. Results

### 2.1. O. octandra Extracts Prevented Body Weight Loss and Normalized Liver Weight

In order to induce continuous liver cirrhosis, TAA multiple intraperitoneal injection (i.p.) strategy was used. Six-week-old male Wistar rats (220–240 g BW) were injected with TAA (100 mg/kg BW) or distilled water (disease control, DC), crude leaves suspension (CLS), boiled leaves extract (BLE), sonicated leaves extract (SLE), methanol leaves extract (MLE) or hexane leaves extract (HLE) (500 mg/kg BW) twice a week. Along with these treatments, we have set healthy controls (HC) by saline i.p. and distilled water gavaging. After 15 weeks, liver and blood samples were collected (Figure 1a).

The endpoint weight and weekly body weight gain were recorded. In comparison with HC (173.6 ± 2.68 g), DC showed significantly low body weight gain (54.8 ± 3.04 g). Compared to rats in DC, the rats in CLS (159.8 ± 2.35 g), BLE (151.2 ± 5.62 g) and SLE (130.2 ± 7.21 g) groups showed significantly higher body weight gains. On the other hand, MLE (65 ± 4.06 g) and HLE (59 ± 4.73 g) did not show any treatment effects (Figure 1b). In the DC group, liver indices (6.05 ± 0.15%) showed a significantly higher liver/body weight percentage compared to the HC (3.04 ± 0.04%). The liver indices of CLS (3.37 ± 0.08%), BLE (3.82 ± 0.07%) and SLE (4.24 ± 0.04%) treatments showed significantly lower values compared to that of DC rats (Figure 1c). Interestingly, CLS, BLE and SLE treatments appear to have the potential to ameliorate TAA-induced liver damage.

### 2.2. O. octandra Extracts Restored Serum Concentrations of Liver Enzymes

To test liver functions, the serum alanine aminotransferase (ALT), aspartate aminotransferase (AST) and alkaline phosphatase (ALP) levels were measured at the end of the treatment period. The DC group showed significantly higher ALT (360.8 ± 6.11 U/L), AST (267 ± 4.13 U/L) and ALP (776.1 ± 5.81 U/L) values compared to HC, which were 156.78 ± 4.86, 118.62 ± 3.17 and 359.6 ± 7.73 U/L, respectively. In comparison to the DC, significantly low serum concentrations of ALT, AST and ALP were observed in the CLS (222.5 ± 1.61, 120.68 ± 1.34 and 370.26 ± 7.79 U/L), BLE (242.88 ± 3.81, 130.92 ± 1.26 and 422.66 ± 18.85 U/L) and SLE (242.86 ± 8.02, 155.4 ± 1.42 and 513.2 ± 46.55 U/L) groups. (Figure 2a–c). The findings confirm the hepatocellular damage in DC as evident by significantly higher enzyme levels. Treatments of CLS, BLE and SLE have shown protection against TAA-induced hepatotoxicity by decreasing intracellular enzyme leakage.

### 2.3. O. octandra Extracts Restored Gross Liver Appearance

The appearance (gross anatomy) of the liver surface was observed, and images were taken at the end of the experiment period to confirm the disease induction and treatment effects of the different leaf extracts of *O. octandra*. HC livers showed a normal liver appearance with a smooth and shiny surface (Figure 3a). Prominent hepatic nodules of variable sizes were seen on the livers of DC (Figure 3b). The CLS treated livers showed smooth surfaces similar to the HC (Figure 3c), while the BLE (Figure 3d) and SLE (Figure 3e) treated livers showed mild irregularity of the surface, although there were no prominent nodules. The livers of MLE (Figure 3f) and HLE (Figure 3g) groups showed prominent nodules with variable sizes similar to DC (Figure 3b). Liver surface appearance changes in CLS, BLE and SLE treatments indicated mild improvements. The appearances further confirmed the recovery and possible protective effect of the extracts. Nevertheless, MLE and HLE did not show any favorable improvement.

### 2.4. O. octandra Extracts Restore Liver Architecture

We further examined the histopathological changes by staining the liver sections with hematoxylin and eosin (HE) and with Masson’s trichrome (MT) for collagen deposition, and micrographs were captured under the light microscope. HC livers showed normal hepatic architecture (Figure 4a and Figure 5a). The livers from DC showed regenerating hepatocytic nodules with complete fibrous bridges in hepatic parenchyma (Figure 4b and Figure 5b). The CLS treated livers showed mild fibrosis around the centrilobular region livers (Figure 4c and Figure 5c). Further, incomplete fibrous bridge formation (Figure 4d and Figure 5d) was observed in the livers of BLE, while complete fibrous bridge formation with regenerating hepatocytic nodules was observed in SLE (Figure 4e and Figure 5e), MLE (Figure 4f and Figure 5f) and HLE (Figure 4g and Figure 5g) treatment receiving rats. The data confirm the establishment of cirrhosis in DC since complete fibrous bridges and hepatocytic nodules were very prominent under the TAA challenge. CLS and BLE treatments showed possible curative effects against the TAA-induced liver cirrhosis. However, complete fibrous bridges and nodule formation could still be seen in SLE, MLE and HLE treated liver sections, which may be due to mild or no beneficial effects.

### 2.5. O. octandra Extracts Ameliorate Liver Fibrosis

The collagen deposition of MT-stained liver sections was analyzed by Image J software. The liver fibrosis was significantly increased in DC compared to that in the HC livers. However, the treatment of CLS, BLE and SLE extracts indicated a significant (*p* < 0.001) reduction of collagen deposition compared to DC. Collagen deposition in MLE and HLE treatments was not reduced and was similar to DC (Figure 6). These results further show possible hepatoprotective effects previously identified in aqueous extracts (CLS, BLE and SLE) but not in MLE and HLE.

### 2.6. O. octandra Extracts Prevent Up-Regulation of Pro-Inflammatory and Profibrotic Cytokine mRNA

To further confirm the disease establishment and the effects of different leaves extract, relative mRNA expressions of tumor necrosis factor-alpha (*Tnf-α*), alpha-smooth muscle actin (*α**-Sma*), transforming growth factor-beta 1 (*Tgf-β1*) and vascular endothelial growth factor 2 (*Vegf-R2*) were measured.

As shown in Figure 7, in comparison to the HC (set as 1.0), significantly higher levels of *Tnf-α* (36.06 ± 0.85), *α**-Sma* (57.22 ± 0.63), *Tgf-β1*(7.39 ± 0.05) and *Vegf-R2* (9.83 ± 0.05) mRNA expressions were observed in DC. the *Tnf-α* expression level was significantly lower in CLS (10.06 ± 0.12), BLE (3.62 ± 0.27) and SLE (9.35 ± 0.64) in comparison to DC (Figure 7a). Moreover, *α-Sma* mRNA expression level in CLS (2.94 ± 0.465), BLE (13.53 ± 0.67) and SLE (23.22 ± 0.52) treated livers was significantly lower than the DC (Figure 7b). *Tgf-β1* mRNA expression level in CLS (1.56 ± 0.05), BLE (3.35 ± 0.15) and SLE (1.90 ± 0.29) were significantly lower than DC (Figure 7c). In addition, *Vegf-R2* mRNA expression levels in CLS (1.27 ± 0.45) and BLE (5.16 ± 0.03) were significantly lower than DC and in SLE (7.18 ± 0.05), livers did not show a significant difference compared to DC (Figure 7d). The data confirmed the establishment of cirrhosis in DC within 15 weeks by significantly increased mRNA expression levels of the identified markers. Significantly reduced expression levels were observed for pro-inflammatory and fibrotic cytokines expressions in CLS, BLE and SLE further confirmed the hepatoprotection against TAA-induced liver cirrhosis at the molecular level as well.

### 2.7. O. octandra Extract Prevents Angiogenesis

The *Vegf-R2* mRNA expressions in CLS and BLE treated rat livers suggested a potent anti-angiogenesis effect. Thus, to further study the effect, a standard in vitro angiogenic assay using human umbilical vein endothelial cells (HUVEC) seeded on a Matrigel model was used. The attachment of the cells occurred in the first hour and then followed by their migration towards each other over the next 2–4 h, forming capillary-like tubes, which matured by 6–16 h. In the control group, the cells were attached, aligned and formed tubes with a lumen that appears as a network (Figure 8a). The addition of BLE to the HUVEC medium significantly inhibited the formation of vessel-like structures and resulted in the cells remaining separated and somewhat rounded as solitary cells (Figure 8a). The HUVEC sprouting was significantly less pronounced with the addition of increasing concentrations of BLE. Furthermore, the number of branch points and tube length was significantly reduced with BLE treatment compared to that of the control (Figure 5b and Figure 8c). These observations further confirm the anti-angiogenesis effect that was more potent in BLE.

## 3. Discussion

Liver cirrhosis and its sequelae of liver failure, portal hypertension and hepatocellular carcinoma are major causes [11] of chronic illnesses leading to deaths worldwide. However, at present, there is no medical treatment for chronic liver disease except liver transplantation, which is hampered by the limited availability of donor organs [12]. It is therefore pertinent to establish the efficacy as well as the safety of medicinal plant preparations through proper toxicological assessments. However, the major drawbacks in formulating therapies with herbal preparations include the lack of data on safety, efficacy and standardization. In addition, studies using medicinal plants were often reported without providing possible molecular or signaling pathways. The medicinal plant *O. octandra* from the family Melastomaceae has been used to treat various liver diseases by indigenous medical practitioners [13]. Thus, the present study was undertaken to provide empirical evidence on the efficacy and safety of *O. octandra* leaf extracts that produced hepatoprotective activity in an animal model of liver cirrhosis.

According to previous research [9,13], the aqueous extracts of *O. octandra* were protective against *tert*-butylhydroperoxide-, acetaminophen-, carbon tetrachloride-(CCl_4_) and galactosamine-induced hepatocyte damage; however, no studies have been performed to reveal the effect of different types of extracts of *O. octandra* in chronic liver injury and cirrhosis.

In the present study, treatments with methanol-derived (MLE) and hexane-derived (HLE) leaf extracts failed to demonstrate positive effects since the treated livers still had altered gross liver appearance and showed a similar nodular appearance to DC. However, in marked contrast, treatments of crude leaf suspension (CLS), boiled (BLE) or sonicated (SLE) leaf extracts maintained a similar gross appearance to those of HC livers, suggesting that CLS, BLE and SLE, but not MLE and HLE, were hepatoprotective in TAA-challenged chronic liver damage. From these results and qualitative analysis of bioactive compounds (Appendix A), it is clearly indicated that the bioactive molecule/s in *O. octandra* might not be simple triterpenoids or polyphenols since methanol and hexane have a great potential to extract those types of bioactive compounds. Thus, it leads us to suspect that a compound or a group of compounds in polyphenols, triterpenoid glycoside or other water-soluble fractions has the potential for liver protection. To find out responsible biomolecules, we are now developing the screening system using *Tnf-α*, *α-Sma*, *Tgf-β1* and *Vegf-R2* expression monitoring during a TAA stress challenge. Moreover, the hepatoprotective effect was supported by the findings of body weight gain and liver index. The treatments of CLS, BLE or SLE were able to normalize the body weights and liver index of TAA-challenged animals showing comparable levels to that of the HC. These findings agree with the previous findings where an aqueous extract of *O. octandra* has been shown to possess hepatoprotective effects [14]. In addition, Thabrew and colleagues have reported that *O. octandra* aqueous leaf preparation has inhibitory effects on both protein and glycogen synthesis in the injured liver [9]. However, our initial toxicology analysis has proven that the doses of CLS, BLE and SLE used in our study were safe without any adverse toxicological outcomes (Appendix A).

The increased release of liver enzymes, such as ALT, AST and ALP, represents chronic liver damage in rats administered with TAA [15]. However, treatment of TAA injected rats with CLS, BLE and SLE extract reduced these enzyme levels to comparable levels of HC animals, suggesting that plant extracts helped to restore liver functions of the damaged liver parenchyma and improved its architecture. The protective effect that leads to the recovery of liver architecture was likely due to the presence of active compounds in the aqueous extracts of *O. octandra* leaves [16,17]. Therefore, the findings of the present study warrant further investigations to identify and characterize active compounds present in the preparations of different extracts of *O. octandra*.

Histopathological observations from MT staining were used to assess the level of fibrosis. Supporting previously published work [7], histological sections from CLS and BLE livers showed close to normal cellular architecture, which corroborates with improved liver biochemistry. These findings indicate that crude suspension or boiled leaf extracts of *O. octandra* could be considered as ideal preparations with the highest hepatoprotective properties.

Treating TAA-challenged cirrhotic rats with CLS, BLE and SLE markedly reduced the expression of pro-inflammatory cytokine *Tnf-α*, which is linked to the secretion of pro-fibrotic cytokines. Indeed, the treatment with the extracts reduced the expression of *Tgf-β1*, a potent pro-fibrotic cytokine secreted by activated hepatic stellate cells (HSCs) [3]. It is well accepted that the perpetuation of inflammation with subsequent activation of pro-fibrotic signals leads to the activation of hepatic stellate cells and subsequent release of excess amounts of extracellular matrix (ECM), resulting in fibrosis [3,18]. Thus, supporting these observations, our data indicate that the plant extracts markedly inhibited the activation of HSCs, as reflected by a reduced expression of HSC marker *α-Sma*. Importantly, the inhibition of HSC activation by *O. octandra* extracts led to a reduced ECM deposition, leading to a profound reduction in liver fibrosis [19]. Furthermore, our findings are in agreement with recent reports that suggested that other herbal plant extracts, such as fucoidan extracted from seaweed preparations, showed anti-fibrotic activity, as fucoidan administration reduced gene expression of *Tgf-β1, Tnf-α* and *α-Sma* in rats with liver disease [20,21]. Overall, the suppressive effects of CLS, BLE and SLE on gene expression of *Tnf-α, α-Sma, Tgf-β1* and subsequent reduction in fibrosis provide strong evidence that the aqueous leaf extracts of *O. octandra* is a novel, anti-fibrotic therapy that has a potential to treat patients with liver fibrosis.

To validate the statistical outcomes, Cohen’s d_s_ and Hedges’s g_s_, which describe the standardized mean differences between two sets of independent measurements based on sample average, were calculated. To estimate the amount of variance accounted for in the sample, Eta squared was calculated. For the estimation of variance accounted in the population, omega squared was calculated. According to the effect size analysis by the above methods, the number of animals used in this study was validated and can be used for future studies that follow this disease induction procedure (Appendix A).

Moreover, recent data have suggested that angiogenesis plays an important role in the development of fibrosis and cirrhosis. Of the two most effective extracts, BLE was used to evaluate the angiogenic effect, whereas CLS could not be used to treat cultured cells due to its particulate nature. The formation of capillary-like tubes in culture by endothelial cells on the basement membrane matrix is a powerful in vitro model to screen various factors that promote or inhibit angiogenesis [22]. In normal endothelial cell cultures treated with BLE, the attached cells remained somewhat rounded solitary cells. However, application of *O. octandra* to endothelial cells markedly inhibited both branching and tube elongation by more than 80% at the highest dose and that this inhibition was well evident even at the lowest dose employed. It is interesting to note that suppression of both branching and elongations of tube structures were dose-dependent, with the highest dose of *O. octandra* extract causing the maximum inhibition. The effects observed with the application of *O. octandra* extract are likely attributable to a reduction of *Vegf* and/or its receptor *Vegf-R2*, as they are considered as potent angiogenic factors and that agents that specifically inhibit angiogenesis are known to alleviate hepatic fibrosis [23]. Indeed, we show that increased *Vegf-R2* expression in DC was markedly inhibited by BLE treatment. In line with this, the findings that CLS and BLE and extracts, which decreased *Vegf-R2* gene expression, confirmed the anti-angiogenic effect of *O. octandra*, leading to improved liver fibrosis in rats with liver cirrhosis.

## 4. Materials and Methods

### 4.1. Plant Materials

Leaves of *O. octandra* were collected from the herbarium of the Department of Animal Science, Faculty of Agriculture, University of Peradeniya, Peradeniya, Sri Lanka. The plant specimens were authenticated by the Curator of the Royal Botanical Gardens in Peradeniya, Sri Lanka, and a voucher specimen (*Osbeckia octandra* specimen No. UB 89) was deposited at the national herbarium.

### 4.2. Preparation of Leaf Extracts

Fresh *O. octandra* leaves (50 g) were washed with distilled water and air-dried at room temperature, and freeze-dried using a freeze-drier (Christ-Alpha 1-4 LD plus, Osterode am Harz, Germany). Several drying cycles were used to obtain the total amount of powder and stored at −20 °C until used. The crude leaf suspension (CLS) was prepared by suspending 1 g of freeze-dried leaf powder in 20 mL of distilled water just before administration. A total of 30 g of freeze-dried powder was mixed with 480 mL of distilled water and then boiled for boiled leaf extract (BLE), and the same mixture was sonicated in VWR Ultrasonic cleaner (USC 1700 D, Randor, PA, USA) for 20 min to obtain sonicated leaf extract (SLE). A total of 30 g of freeze-dried powder, mixed with 480 mL of methanol or hexane [24] and sonicated for 20 min, followed by filtration and solvent evaporation using a rotary evaporator (Heidolph, Schwabach, Germany) at 50 °C prior to overnight (37 °C) vacuum drying in an oven (Heracus, Schwabach, Germany) was used for methanol leaf extract (MLE) and hexane leaf extract (HLE). Prepared powder (BLE, SLE, MLE and HLE) (1 g) from each extract was dissolved in 20 mL of distilled water to make a stock solution just before administration.

### 4.3. Experimental Animals

Thirty-five, six-week-old, male Wistar rats (220–240 g BW) were obtained from the Medical Research Institute, Borella, Sri Lanka. They were housed individually at the vivarium of the Faculty of Medicine, University of Peradeniya, Peradeniya, Sri Lanka under standard conditions (22 ± 3 °C with a 12 h light-dark cycle). The animals were fed with a standard commercial diet and provided tap water *ad libitum*. Thirty rats were given an injection of TAA (Sigma, St. Louis, MO, USA) dissolved in physiological saline (0.9% NaCl) at a dose of 100 mg/kg body weight, intraperitoneally, twice a week, for 15 weeks to induce cirrhosis [25]. The TAA injected rats were divided into six groups (*n* = 5 per group). The first group of rats who served as the TAA only treated group and orally gavaged twice weekly with distilled water (TAA only group as DC). The remaining rats were orally gavaged twice weekly with CLS (500 mg/kg BW), BLE (500 mg/kg BW), SLE (500 mg/kg BW), MLE (500 mg/kg BW) and HLE (500 mg/kg BW). Leaf extracts and distilled water treatments were done simultaneously with TAA injection to investigate the protective effects of the different extracts. The doses used in this study were proven to be non-toxic (the toxicology experiment data are shown in Appendix A). The remaining five rats were injected intraperitoneally with an equivalent volume of normal saline and orally gavaged with an equivalent volume of distilled water and served as the control group (HC). All rats were euthanized under isoflurane anesthesia [26] in an induction chamber with isoflurane 3.5% for 2–5 min at 15 weeks after the treatments, and blood and liver samples were collected. Experimental procedures were carried out according to the institutional animal ethics guidelines on the conduct of animal experimentation and animal care (Certificate No. VER-16-001).

### 4.4. Body Weight Assessment and Liver Weight Assessment

The body weights of rats were measured at the beginning of the study, and the final body weights were measured before sacrificing the animals. The liver weights were taken after sacrificing the animals at 15 weeks with a pocket scale (Pocket scale, B-01, Zhejiang, China).

### 4.5. Serum Collection and Blood Chemistry

Serum was separated from collected blood and stored at −20 °C. The samples were subjected to the analysis of alanine aminotransferase (ALT), aspartate aminotransferase (AST) and alkaline phosphatase (ALP) using a semi-automatic analyzer (3000 Evolution, Milan, Italy) with commercial kits following the manufacturer’s instructions (Randox, County Antrim, UK).

### 4.6. Tissue Preparation and Histopathology

Collected liver samples were fixed, embedded and sectioned using a microtome and mounted on glass slides. Liver sections were stained with hematoxylin and eosin (HE) for histopathology and Masson’s trichrome (MT) for collagen deposition [27].

### 4.7. RNA Extraction and Real-Time Quantitative Polymerase Chain Reaction (qPCR)

Liver samples from the median lobe were immediately soaked in RNAlater (Sigma) and were stored at −80 °C. Real-time qPCR was performed as described previously [25] and primers used are shown in Appendix A. The *18S* mRNA expression was used as the internal control to normalize mRNA expression data.

### 4.8. Assessment of the Effects of Leaf Extract on Angiogenesis In Vitro

The effect of BLE on endothelial cell sprouting was studied using human umbilical vein endothelial cells (HUVECs) cultured on the basement membrane matrix (Matrigel, BD Biosciences, Franklin Lakes, NJ, USA) as described previously (*n* = 3) [28].

### 4.9. Statistical Analysis

Data were expressed as mean ± standard error of the mean (SEM) and analyzed using ANOVA. Multiple comparisons of means were performed using Dunnett’s test using Minitab (version 17 for Windows, Minitab Ltd., Coventry, UK). A probability value of *p* < 0.05 and *p* < 0.01 was considered a significant change. To validate the statistical outcomes, the effect sizes were estimated by using Cohen’s d_s_, Hedges’s g_s_, Eta squared and Omega squared. (All effect size data are shown in Appendix A). Statistical analysis for effect sizes was done using IBM SPSS and Microsoft Excel.

## 5. Conclusions

The present study demonstrates that CLS, BLE and SLE of *O. octandra* ameliorate liver fibrosis in cirrhotic rats by inhibiting pro-inflammatory and pro-fibrotic cytokine secretion and angiogenesis. We conclude that extracts of the medicinal plant *O. octandra* have the potential for use as a treatment of liver fibrosis/cirrhosis. However, further investigations are required to identify active compounds present in *O. octandra* extracts that render anti-fibrotic properties, opening new avenues to develop novel therapeutic agents to treat cirrhotic patients.

## Figures and Tables

**Figure 1 molecules-26-04836-f001:**
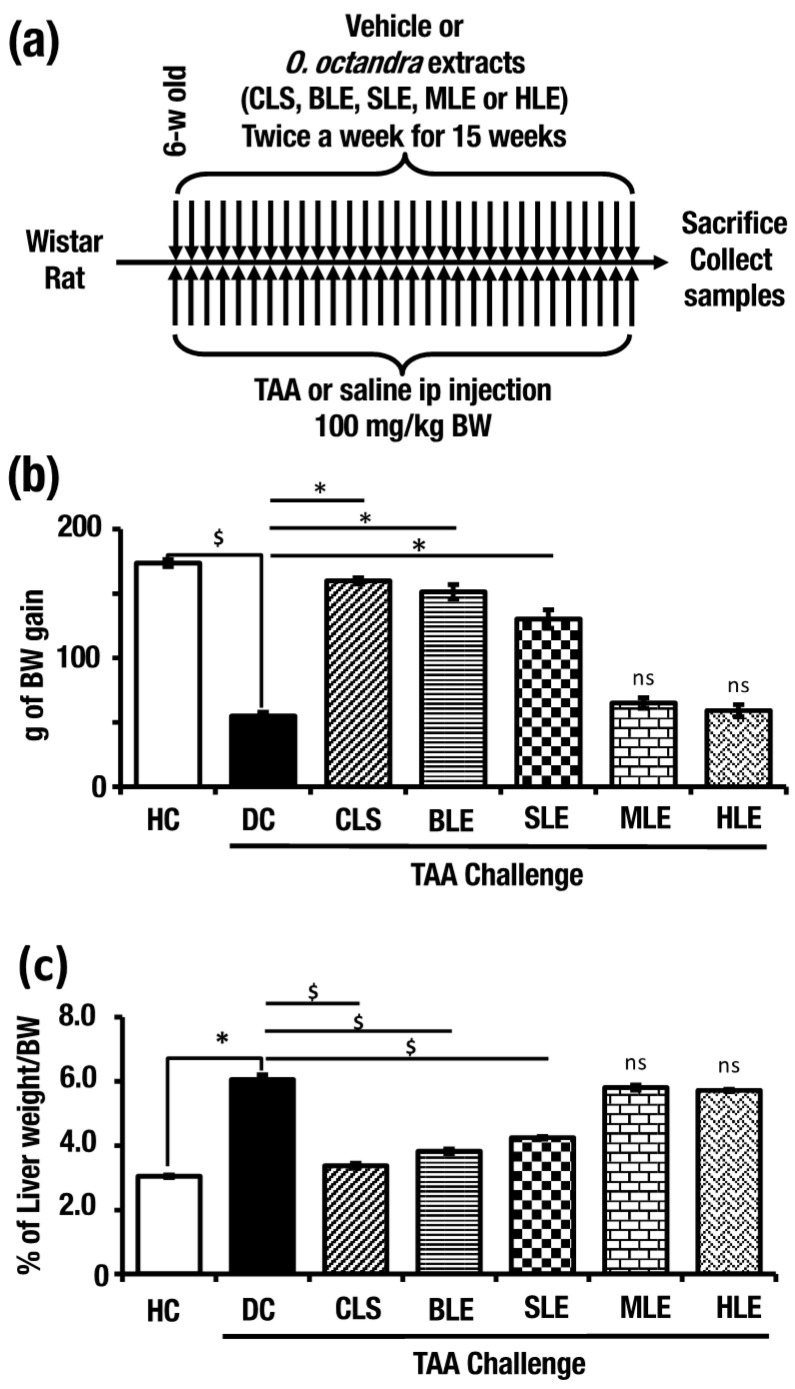
Aqueous extracts from *O.*
*octandra* prevents body and liver weight retardation during the TAA challenge. (**a**) Experimental procedure. (**b**) Body weight gain from the beginning to the end from each of the seven groups of rats (HC, DC CLS, BLE, SLE, MLE or HLE treatment model) during the TAA challenge. The values are indicated in grams of increase/decrease using the starting weight. (**c**) Liver/body weight ratio at the end from each of the seven groups of rats. Error bars represent standard errors of the means (SEM) in each group (*n* = 5). The statistical significance has been indicated in two ways; (i) disease induction effect during the TAA challenge (HC vs. DC), (ii) *O.*
*octandra* treatment consequence (DC vs. CLS, BLE, SLE, MLE and HLE), with one-way ANOVA followed by the Dunnett’s test. $, significant decrease; $, *p* = 0.05 to 0.01. *, significant increase; *, *p* = 0.05 to 0.01. ns, *p* > 0.05.

**Figure 2 molecules-26-04836-f002:**
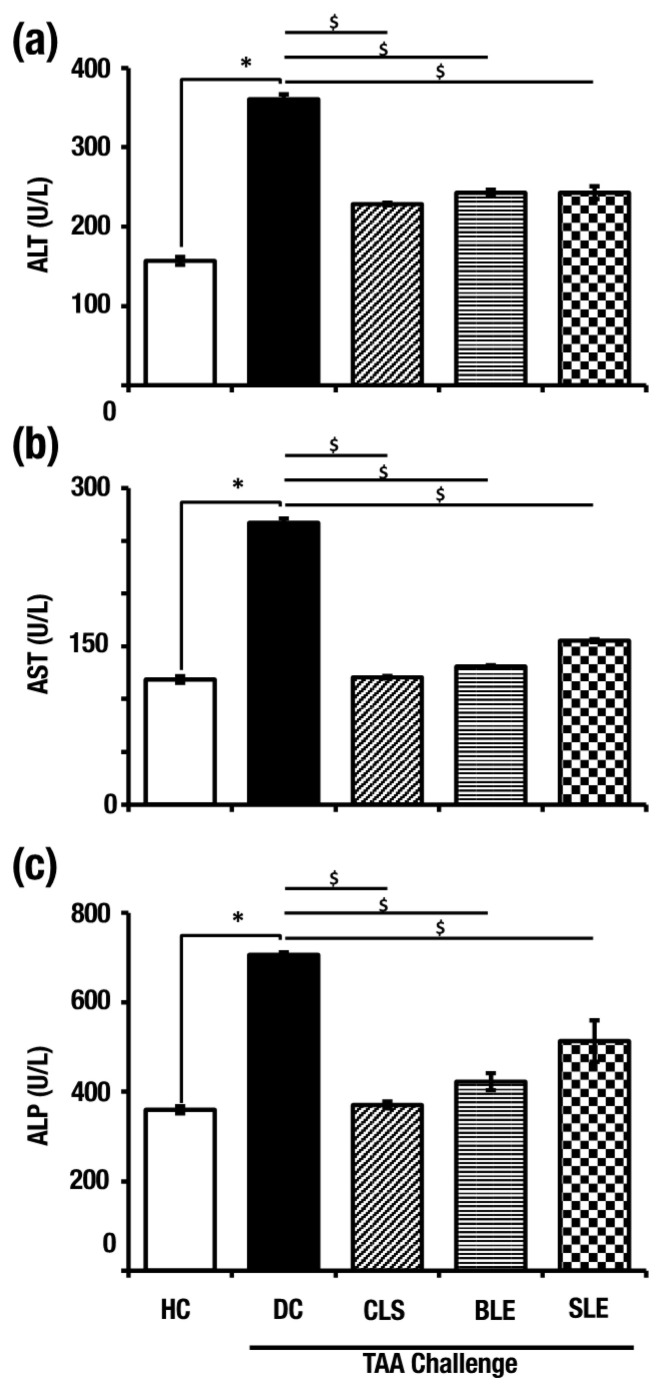
Selected aqueous extracts (CLS, BLE and SLE) protect the liver from TAA-induced damage by reducing the leakage of liver enzymes at the end of the treatment period. (**a**) Serum ALT levels, (**b**) serum AST levels and (**c**) serum ALP levels of HC, DC and after treatments of aqueous extracts along with TAA challenge. Error bars represent standard errors of the mean (SEM) in each group (*n* = 5). The statistical significance of results has been shown in two ways; (i) disease induction effect during the TAA challenge (HC vs. DC), (ii) *O.*
*octandra* treatment effect (DC vs. CLS, BLE and SLE) was analyzed with one-way ANOVA followed by the Dunnett’s test. $, significant decrease; $, *p* = 0.05 to 0.01. *, significant increase; *, *p* = 0.05 to 0.01.

**Figure 3 molecules-26-04836-f003:**
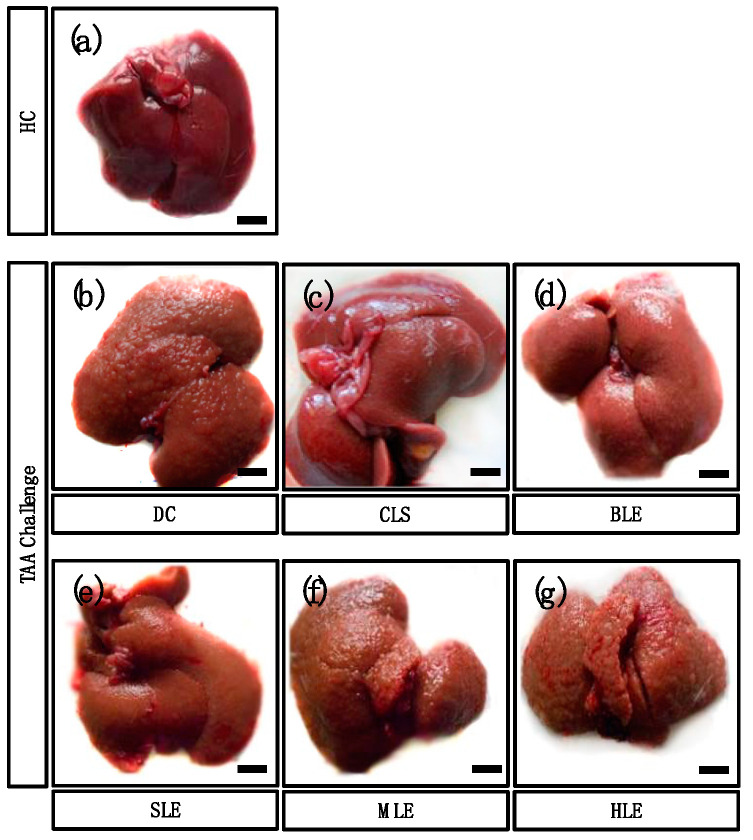
Aqueous extracts (CLS, BLE and SLE) demonstrated more potency to protect the liver against TAA challenge. (**a**) HC livers showed a normal appearance with a smooth surface. (**b**) Numerous nodules of various sizes can be seen on the liver surface of DC. (**c**) Livers treated with CLS showed a smooth surface. (**d**) Small nodules can be seen on the surfaces of livers treated with BLE and (**e**) SLE. (**f**,**g**) Nodules of variable sizes were present on the surfaces of livers treated with MLE and HLE. Bar = 10 mm.

**Figure 4 molecules-26-04836-f004:**
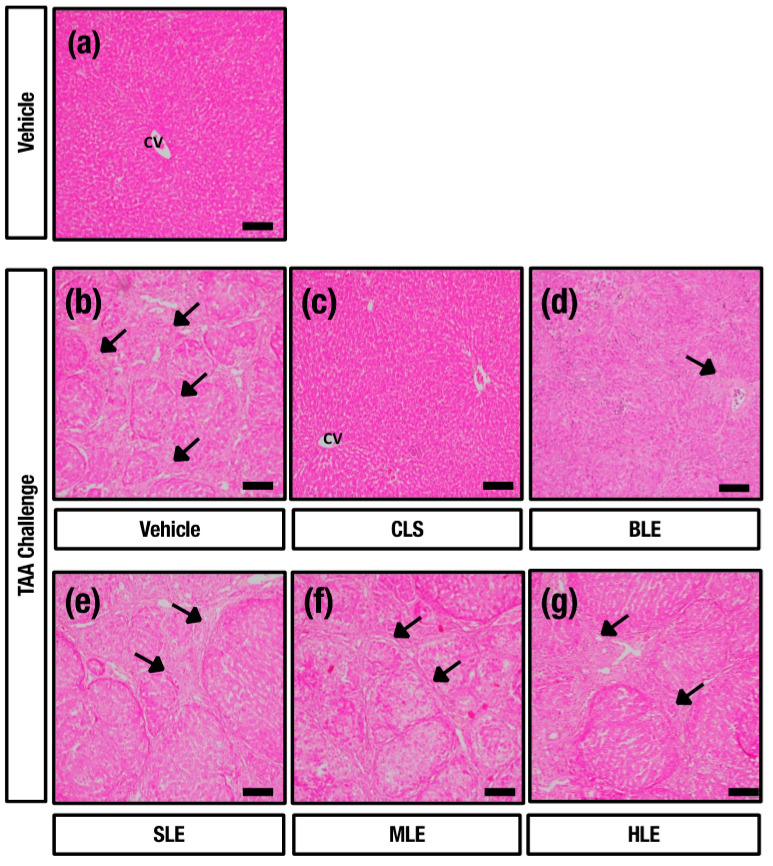
Hepatoprotective effect of aqueous extracts (CLS and BLE) against TAA-induced liver cirrhosis. Histopathological sections of livers (stained with HE) from HC, DC and different leaf extract treatments of *O. octandra* CLS, BLE and SLE) after 15 weeks. (**a**) HC livers showed normal hepatic architecture and without histopathological changes. (**b**) Regenerating hepatic nodules formed by complete fibrous bridges were seen in the livers of DC. (**c**) Animals treated with CLS showed normal hepatic architecture. (**d**) Incomplete fibrous bridge formation was seen in the animals treated with BLE. (**e**–**g**) Complete fibrous bridge formation with regenerating hepatocytes was observed in animals treated with SLE, MLE and HLE, respectively. CV, central vein. Bar = 100 µm.

**Figure 5 molecules-26-04836-f005:**
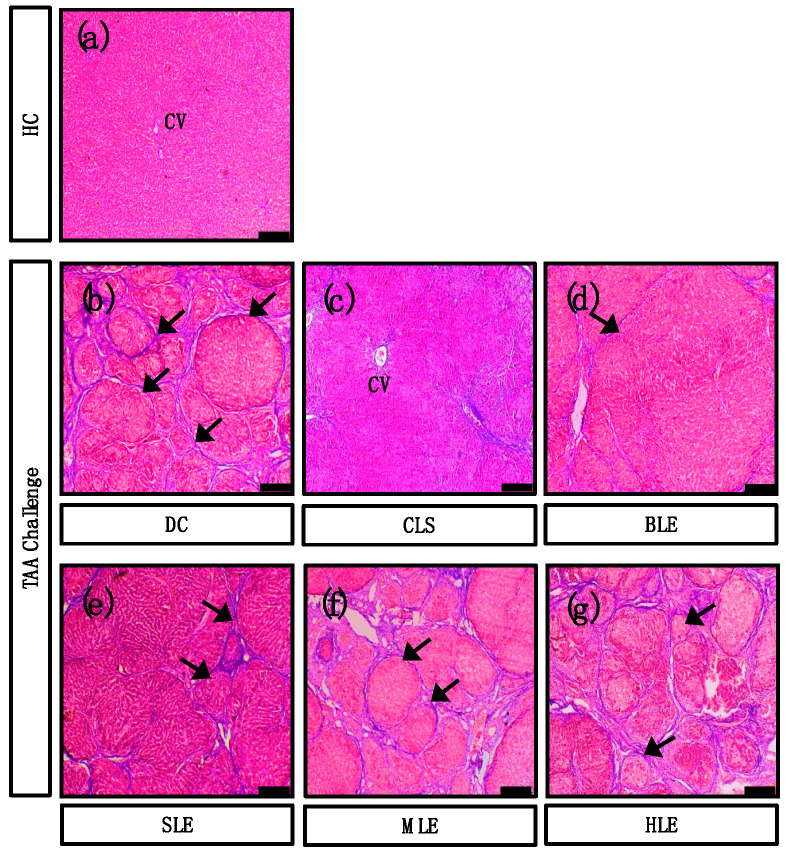
Aqueous extracts (CLS and BLE) showed an anti-fibrotic effect against TAA-induced liver cirrhosis. Histopathology of (stained with Masson’s trichrome) HC, DC and different leaves extracts of *O. octandra* treated CLS, BLE and SLE) livers at 15 weeks. (**a**) In the HC livers, collagen fibers were mainly seen in the vascular wall. (**b**) Increased collagen deposition in the thickened fibrous bridges was seen in DC. (**c**) A small amount of collagen was detected mainly in the periportal and centrilobular areas in livers treated with CLS. (**d**) Livers treated with BLE showed a few collagen deposits as thin fibrous bridges. (**e**–**g**) Increased amounts of collagen with thickened fibrous bridges were observed in livers treated with SLE, MLE and HLE. CV, central vein. Arrows indicate collagen depositions. Bar = 100 µm.

**Figure 6 molecules-26-04836-f006:**
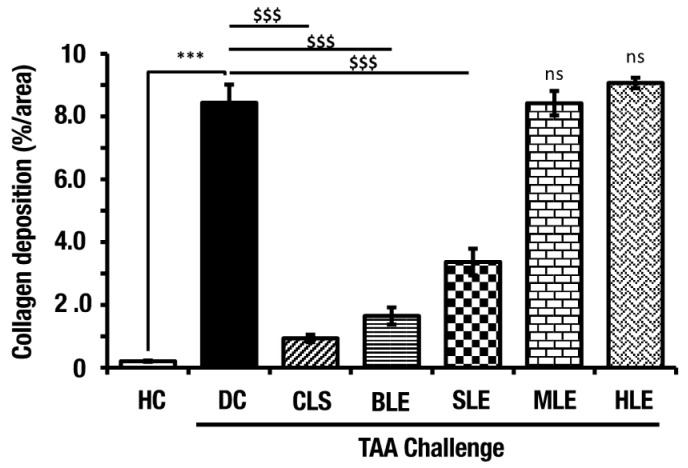
Treatments with aqueous extracts (CLS, BLE and SLE) showed a minimum amount of collagen deposits, indicating their hepatoprotective activity in TAA-induced liver cirrhosis. MT-stained liver sections were quantified using image J software. The values are indicated as percentages of the total surface. Error bars represent standard errors of the means (SEM) in each group (*n* = 5). The statistical significances are shown in two ways; (i) disease induction effect during the TAA challenge (HC vs. DC), (ii) *O.*
*octandra* treatment consequence (DC vs. CLS, BLE, SLE, MLE and HLE), with one-way ANOVA followed by the Dunnett’s test. $, significant decrease; $$$, *p* < 0.001. *, significant increase; ***, *p* < 0.001. ns, *p* > 0.05.

**Figure 7 molecules-26-04836-f007:**
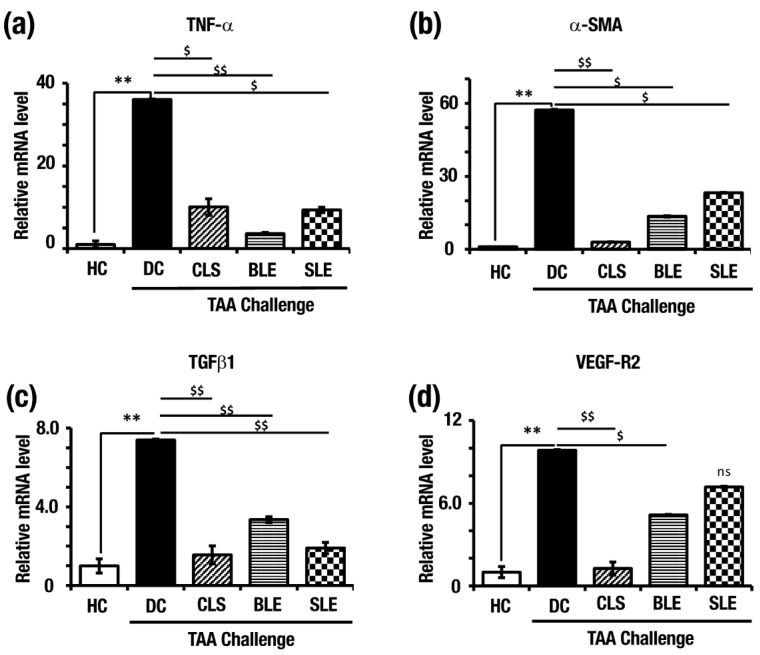
mRNA expressions confirmed the establishment of liver cirrhosis in DC and the hepatoprotective effect of aqueous extracts in the other treatments. (**a**) *Tnf-α*, (**b**) *α-Sma*, (**c**) *Tgf-β1* and (**d**) *Vegf-R2* mRNA expressions determined by qPCR. Expression levels are normalized to *18S* mRNA level. Error bars represent standard errors of the means (SEM) in each group (n = 5). The statistical significances are indicated in two ways; (i) disease induction effect during the TAA challenge (HC vs. DC), (ii) *O.*
*octandra* treatment consequence (DC vs. CLS, BLE and SLE), with one-way ANOVA followed by the Dunnett’s test. $, significant decrease; $, *p* = 0.01 to 0.05, $$, *p* = 0.001 to 0.01, *, significant increase; **, *p* = 0.001 to 0.01. ns, *p* > 0.05.

**Figure 8 molecules-26-04836-f008:**
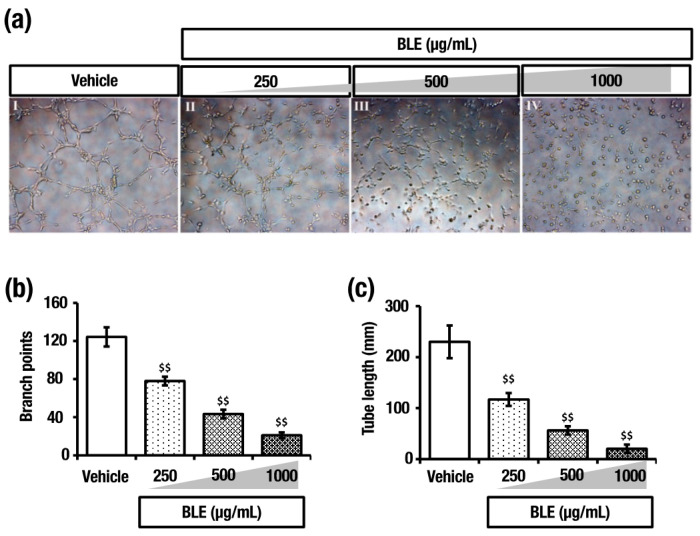
Boiled leaf extract (BLE) of *O. octandra* inhibited angiogenesis in the HUVEC model. (**a**) Representative micrographs of 4 h post-seeding. (I) HUVECS were attached, aligned and organized in tubes with a lumen that appeared as a network when cultured on ECM. (II–IV) In the presence of *O. octandra* extract, HUVECs were attached but remained somewhat rounded as solitary cells. The effect of *O. octandra* was dose-dependent, and when the concentration increased from 250 to 1000 µg/mL, branching and tube formation was significantly inhibited. (**b**) Cumulative tube length and (**c**) number of branch points of capillary-like structures were also significantly reduced at 4 h. Error bars represent the standard error of the means (SEM) in each group (*n* = 3). The statistical significance was calculated with one-way ANOVA followed by Dunnett’s test. $, significant decrease; $$, *p* = 0.001 to 0.01.

## Data Availability

Not applicable.

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
