# Peer review of "Anti-Fibrotic and Anti-Angiogenic Activities of Osbeckia octandra Leaf Extracts in Thioacetamide-Induced Experimental Liver Cirrhosis"

_molecules, 2021, doi:10.3390/molecules26164836_

Round 1
Reviewer 1 Report
- In the study were used different type of extract by O. octandra and their biological activities were adequately described. Instead, serious lake of information was related to the molecular description of these preparations. Firstly, which is the hypnotized biological active molecule, and which was its concentrations in the formulations tested?
Author Response
Dear Reviewer,
Please see the attachment.
Sincelery,
Tadayuki Tsujita Ph.D

Reviewer 2 Report
The Authors of the manuscript have tried to assess an anti-angiogenic properties of Osbeckia octandra in the cirrhotic rat model. They tested crude leaf suspension, boiled leaf extract, sonicated leaf extract, methanol leaf extract and hexane leaf extract. The Authors conclude that extracts of the medicinal plant Osbeckia octandra has potential for use as a treatment of liver cirrhosis. This paper is a significant contribution to the scientific discussion about hepatoprotective and anti-angiogenic effect of Osbeckia octandra. It has good scientific quality. Text editing and minor language revisions should be made.
Author Response

(The authors gave the same response as above.)

Reviewer 3 Report
Overview:
The present study aims at providing insight on the potential of traditional medical preparations from O. octandra to alleviate chronic liver damage.
Overall, I believe that the paper is well written and the methods are carefully described. The objectives and the design of the study are clearly stated and identified in the Introduction section, which builds towards the investigation of the hepatoprotective effects of O. octandra extracts on thioacetamide (TAA) induced cirrhosis in rats.
Given the assessment of various biochemical markers (bodyweight, liver enzymes, anatomy and architecture, expression of pro-inflammatory and profibrotic cytokine), the current study informs on the suitability of the herbal preparations to attenuate chronic liver damage. Also, with limited data available, combined with the straightforward findings, the topic of the paper is relevant and of interest for the readers of the journal.
The paper is well structured throughout, and the conclusions are supported by the results. The design and presentation of the research are acceptable. Additionally, the importance of the study is pointed out through its applicability, which stems from the potential of the herbal preparations to exert their hepatoprotective effects in humans. On the other hand, even though the authors provided the interpretation of the obtained results in a well written manner, with the inclusion of appropriate statistics throughout the manuscript, I am unconvinced that the low number of animals is representative for the observed effects. In this context, the conclusions drawn by the authors may be slightly overreaching. This observation is detailed below. Also, some edits are needed to improve the quality of this manuscript.
Therefore, I recommend undergoing major revision.
Major comments:
Abstract: The abstract lacks figures/numbers - the authors should consider adding the information showing the extent of the most important observed changes. For example, what were the values of improved blood markers, gene expression? …
Statistical analysis: Taking into account the nature of the present trial, which consists of sacrificing the subjects (rats), the lowest possible number of experimental units/animals is generally considered to be the adequate choice - Ethical Guidelines for the Use of Animals in Research (https://www.mdpi.com/journal/molecules/instructions#ethics)
However, given the low number of animals used in this study, presenting the associated effect sizes is most useful because they facilitate cumulative science. Effect sizes can be used to determine the sample size for follow-up studies, or examining effects across studies.
Therefore, I suggest the authors reported the effect sizes corresponding to the statistical analysis by calculating the following: Cohen’s dS, Hedges’s gS, Eta squared and Omega squared.
Discussion: The study lacks the assessment of bioactive compounds profiles of the plant extracts. I suggest including several additional information (available literature) with regard to the potential effects of the active compounds (or at least the main classes of compounds) that are present in the aqueous extracts of O. octandra leaves.
Minor comment:
Figure legends: except the explanation of Figure 1, all the rest are quite erratic and confusing – the authors use numbering elements (letters) at the end of the associated section. Please consider using the same arrangement as the one given in the legend of Figure 1 (before), e.g. “(a) Experimental scheme ….; (b) Body weight gain …..” - the same comment applies for the legend of supplementary figures.
Author Response

(The authors gave the same response as above.)

Round 2
Reviewer 1 Report
Authors have properly addessed reviewer's comments. Publication of the present paper is recommended.
Reviewer 3 Report
Overview:
Overall, the manuscript improved significantly.
The updated abstract now provides a clearer idea of the presented study.
The updated figure captions now follow a parallel structure.
Also, I believe the addition of the effect sizes provides a better interpretation of the obtained results.
As a side note, the minus results you have obtained is a result of subtracting the larger mean from the smaller mean in calculating dS, for example. If you reverse the order, subtracting the smaller from the larger, you will obtain the same value, without the minus sign. The use of positive/negative values when reporting effect sizes is somewhat controversial, thus, the result is generally treated as the absolute value of the effect. Nonetheless, the interpretation of magnitude of effect is the same.
As long as the order of factors is appropriately given in Table S3, I believe the reported effect sizes are sufficiently informative when interpreting the results as a whole.
Therefore, I believe the manuscript has been sufficiently improved to warrant publication in Molecules.